# Validation of StepTest4all for Assessing Cardiovascular Capacity in Young Adults

**DOI:** 10.3390/ijerph191811274

**Published:** 2022-09-08

**Authors:** José A. Bragada, Raul F. Bartolomeu, Pedro M. Rodrigues, Pedro M. Magalhães, João P. Bragada, Jorge E. Morais

**Affiliations:** 1Department of Sport Sciences, Instituto Politécnico de Bragança, 5300-253 Bragança, Portugal; 2Research Centre in Sports, Health and Human Development (CIDESD), 5001-801 Vila Real, Portugal; 3Department of Sport Sciences, Instituto Politécnico da Guarda, 6300-559 Guarda, Portugal; 4North East Local Health Unit—Health Care Unit of Santa Maria, 5301-852 Bragança, Portugal

**Keywords:** heart rate recovery, exercise and health, StepTest4all, cardiovascular capacity, cardiovascular classification, validation, VO_2max_

## Abstract

Background: Cardiovascular capacity, expressed as maximal oxygen uptake (VO_2max_), is a strong predictor of health and fitness and is considered a key measure of physiological function in the healthy adult population. The purpose of this study was to validate a specific step test (StepTest4all) as an adequate procedure to estimate cardiovascular capacity in young adults. Methods: The sample was composed of 56 participants, including 19 women (aged 21.05 ± 2.39 years, body mass = 57.50 ± 6.64 kg, height = 1.62 ± 0.05 m, body mass index = 22.00 ± 2.92 kg/m^2^) and 37 men (aged 22.05 ± 3.14 years, body mass = 72.50 ± 7.73 kg, height = 1.76 ± 0.07 m, body mass index = 23.34 ± 2.17 kg/m^2^). Participants were included in one of the following groups: (i) the group used to predict the VO_2max_, and (ii) the group used to validate the prediction model. All participants performed the StepTest4all protocol. The step height and the intensity of the effort was determined individually. Heart rate and oxygen uptake were measured continuously during rest, effort, and recovery phases. The validation process included the following three stages: (i) mean data comparison, (ii) simple linear regression, and (iii) Bland–Altman analysis. Results: The linear regression retained, as significant predictors of the VO_2max_, sex (*p* < 0.001) and heart rate recovery for one minute (*p* = 0.003). The prediction equation revealed a high relationship between measurements (R^2^ = 63.0%, SEE = 5.58). The validation procedure revealed non-significant differences (*p* > 0.05) between the measured and estimated maximal oxygen uptake, high relationship (R^2^ = 63.3%), and high agreement with Bland–Altman plots. Thus, VO_2max_ can be estimated with the formula: VO_2max_ = 22 + 0.3 · (HRR_1min_) + 12 · (sex), where HRR_1min_ is the magnitude of the HR decrease (bpm) in one minute immediately after the step was stopped, and sex: men = 1, women = 0. Conclusions: The StepTest4all is an adequate procedure to estimate cardiovascular capacity, expressed as VO_2max_, in young adults. In addition, it is possible to determine the qualitative level of cardiovascular capacity from the heart rate recovery for one minute, more specifically, poor: <20, moderate: 20 to 34, good: 35 to 49, and excellent: ≥50. This procedure has the benefit of being simple to apply and can be used by everyone, even at home, without specialist supervision.

## 1. Introduction

Cardiovascular capacity, expressed as VO_2max_, is a strong predictor of fitness and has been considered a key physiological measure in the healthy adult population [1]. Testing and maintaining good cardiorespiratory fitness could help to prevent declines in health status within the general population [2].

Exercise effort testing is a procedure that can be used to estimate cardiovascular fitness, diagnose cardiovascular diseases, and predict mortality associated with cardiovascular problems [3,4]. Studies have demonstrated that exercise testing data have powerful prognostic properties, particularly when focused on functional capacity and exercise heart rate dynamics, such as heart rate recovery (HRR) [5,6].

The heart rate (HR) increases with exercise intensity to respond to greater metabolic demand of muscles and other tissues. This increase in HR is tightly regulated by the action of central and peripheral mechanisms that project afferent inputs to medullary centers in the brain. These afferent inputs result in an appropriate efferent response of the autonomic nervous system branches, i.e., a decrease in parasympathetic and an increase in sympathetic activity [7]. After the exercise ends, there is a progressive reduction in metabolic activity and, consequently, HR also decreases. A slow HRR after exercise is a strong indicator of cardiac autonomic dysfunction [7,8]. A delayed decrease in HR during the first minute after graded exercise is a powerful predictor of overall mortality independent of workload, the presence or absence of myocardial perfusion defects, and changes in HR during exercise [1,7,9]. Better levels of aerobic fitness act beneficially on the autonomic control of post-exercise HR, preserving the vagal reentry velocity in healthy middle-aged adults [10]. Thus, there is scientific evidence to accept the association between HRR and prognosis of cardiovascular disease, which supports the recommendation of recording HRR for risk assessment in clinical practice, as a routine [11,12].

Among the variety of tests described for this purpose, the step tests are the most accessible to the general population and have been used for many years [13,14,15]. Step tests can be considered of major importance, as they allow the assessment of cardiovascular capacity as well as identification of the dynamics of HR during effort and in the recovery period [16]. The HR decrease during recovery, and in the period immediately after the end of the exercise can provide a simple, effective, and ecologically valid method of submaximal assessment of cardiorespiratory fitness that can be implemented in a variety of situations [14]. Heart rate recovery can be defined as the reduction in heart rate immediately after the cessation of exercise, usually for one minute [8,16,17].

The use of step tests may be appropriate for the following reasons: (i) they can be used to monitor training status over time [18], (ii) they can estimate the cardiovascular capacity from the HR recovery [19,20,21], and (iii) they allow an appreciation of the functionality of the sympathetic and parasympathetic nervous systems [22]. Thus, its use makes sense both for the assessment of physical capacity, and in clinical practice. Additionally, step tests have the advantage of requiring minimal and portable equipment and marginal space compared to tests utilizing treadmills, shuttle walks, or cycle ergometers. However, most of the existing step tests have one or more of the following limitations: (i) pre-established durations; (ii) frequently eliciting efforts above 85% of maximal oxygen uptake (VO_2max_), which are not recommended for sedentary or elderly people [23]; and (iii) a fixed step height, which is often unsuited for people with short stature, excess weight, or poor physical capacity.

Due to all these limitations, we developed a new step test procedure, StepTest4all, that solves those constraints and can also be adjusted to individual capacity. In StepTest4all, the step height, duration, and level of difficulty can be adjusted to each person regardless of their age, weight, height, or level of physical fitness. Thus, the purpose of this study was to validate the StepTest4all as an adequate procedure to estimate cardiovascular capacity in young adults.

## 2. Material and Methods

### 2.1. Participants and Samples

The sample was composed of 56 participants, including 19 women (aged 21.05 ± 2.39 years, body = mass 57.50 ± 6.64 kg, height = 1.62 ± 0.05 m, body mass index = 22.00 ± 2.92 kg/m^2^) and 37 men (aged 22.05 ± 3.14 years, body mass = 72.50 ± 7.73 kg, height = 1.76 ± 0.07 m, body mass index = 23.34 ± 2.17 kg/m^2^). The sample recruitment excluded people who had any physical limitation that made it impossible for them to go up and down the step, or any other medical contraindication to performing moderate physical effort. Participants signed an informed consent form. All procedures were in accordance with the Declaration of Helsinki regarding human research, and the Polytechnic Ethics Board approved the research design.

Participants were included in one of the following groups: (i) the group for VO_2max_ prediction equation (equation group—EG), and (ii) the group for validation of the prediction model (Validation group—VG). The participants’ characteristics are presented in Table 1.

### 2.2. Data Collection

Anthropometric variables were measured using a digital stadiometer (Seca 242, Hamburg, Germany) and an electronic scale (Seca 884, Hamburg, Germany). VO_2_ and HR were measured using a stationary breath-by-breath electronic metabolic device (Cortex, Model MetaLyzer 3B, Leipzig, Germany). The device includes a heart rate transmitter (Polar Electro Oy, Kempele, Finland). The apparatus was calibrated with standard gases before each test. According to the manufacturer’s manual, the standard error is 0.1% for oxygen and carbon dioxide sensors.

The VO_2_ and HR were measured continuously for each participant while they performed the following activities in sequence: rest, StepTest4all protocol, and recovery. The HR and VO_2_ values obtained as follows, were considered for further analysis: resting values—average of the last minute of resting period, values obtained during StepTest4all—the average values obtained in the last 5 s of each intensity level, recovery phase—the average values obtained in the last 5 s of the first minute of recovery. Resting VO_2_ and resting HR were collected continuously during 10 min sitting on a chair in a silent and dimly lit room. Participants were not allowed to sleep. The values obtained in the last minute were used for data analysis. During the recovery phase, although the HR values were recorded after the first two minutes, we only considered the value after the first minute. We chose one minute recovery time because it is the usually chosen time and has a higher reproducibility [24].

### 2.3. StepTest4all Protocol

For the StepTest4all protocol, each participant performed a continuous progressive test that consisted of going up and down on a stable step. After the step-up phase, the opposite leg also stepped up to the platform, so the participant stood vertically, supported by both legs, before beginning the step-down phase. The step-down phase began with the same leg as the previous phase and ended when the participant was at the beginning point, also standing in a vertical position, supported by both legs.

The step height was calculated for each participant individually, based on characteristics of their cardiovascular capacity. The variables chosen were sex, age, physical fitness, height, body mass index (BMI), and smoking status. For each variable, a numerical value was assigned as follows: (i) sex (women = 0.5; men = 1), (ii) age (senior = 0, adult = 0.5, young = 1), (iii) physical fitness (sedentary = 0, active = 0.5, very active = 1, athlete = 1.5), (iv) body mass index (BMI < 25 = 0.5, BMI < 30 = 0, BMI ≥ 30 = −0.5), and (v) smoking status (smoker = 0, non-smoker = 0.5). From these data, the step height was calculated using the formula: step height (cm) = 4 × sum of those variables + 15, and it could range from 15 to 40 cm.

This formula was the result of multiple tests on people with different characteristics and of different physical ability. The height of the step, although important, may vary slightly, because the control of the load progression intensity until the desired value will be done mainly by increasing the ascent and descent pace.

A height of 40 to 45 cm has been used in other protocols, for example in the Harvard step test (see: https://www.brianmac.co.uk/havard.html (accessed on 28 July 2022)) [25]. In the present study, the step height of 40 cm combined with a high pace, led to an intensity of 80% of the estimated HR_max_ in 5–10 min. This occurs even in subjects with good physical ability and tall stature.

The test started at a rate of 15 cycles per minute (0.25 Hz), controlled by a metronome. In each cycle, the participant went up and down the step, so the cycle ended at the same time the second leg reached the floor. Every minute, the cadence was raised by 2.5 cycles per minute. The maximum expected duration of the test was 10 min.

The lower limit of 15 cycles per minute is a very slow ascent and descent pace that can be performed by anyone and serves as a warm-up. The upper limit of 37.5 cycles per minute is a pace that can only be performed by subjects of high physical capacity.

The test ended when one of the following criteria was met: (i) the HR reached 80% of the maximal heart rate (HR_max_), (ii) when asked by the participant upon feeling uncomfortable with the effort, or (iii) when the participant was unable to perform the exercise at the correct cadence. Immediately after the end of the test, the participants were to stand for two minutes. Although a standing position was mandatory, the participants were encouraged to be at ease, in order to recover as much as possible, and were not permitted to talk, grab, or hold onto anything.

The combination of the step’s height with the rhythm’s increments, associated with a limitation in intensity to finish the test (80% of HR_max_), made it possible to achieve the test objectives in a reasonable amount of time (5 to 10 min) on all kinds of people. In the recovery phase the HR was collected continuously, also with the Garmin Fenix 6, and its HR belt (Garmin International, Inc., Olathe, KS, USA).

The HR_max_ and VO_2max_ determination were made as follows: HR_max_ was estimated according to the formula: HR_max_ = 208 − 0.7 × age [26].

The VO_2max_ was estimated from the individual equation of the regression line associating HR–VO_2_, obtained from the rest data and during 3 or more steps of StepTest4all, more precisely, by calculating the value of VO_2_ corresponding to HR_max_ [27]. We assumed this value as VO_2max_ measured. Individual linear regressions (R^2^) ranged between 0.97 and 0.99 (almost perfect relationship). This is a usual and suitable procedure for evaluating VO_2max_ in subjects for whom a maximal test to exhaustion may have some inconvenience.

The use of submaximal tests to estimate VO_2max_ from the association of HR–VO_2_ is recurrent and has been shown to be adequate [28,29]. Evans and co-workers [30], in a systematic review, found that non-significant differences were reported between the measured and predicted VO_2max_ in 28 equations. The variable most used in the predictive equations was the HR (N = 19). Additionally, Bennett and co-workers [31] validated a submaximal treadmill-based protocol. The authors noted that VO_2max_ was better estimated when calculated from the projection of HR_max_ [31].

With the participants of the present study, it would not be appropriate to perform maximal testing to exhaustion. They were categorized either as sedentary or active people, that is, they either did not practice any physical activity or did one or two sessions of recreative exercise, such as walking or cycling.

### 2.4. Statistical Analysis

Initially, Kolmogorov-Smirnov and Levene tests assessed normality and homoscedasticity, respectively. Descriptive data means and one standard deviation (1SD) were calculated. For the prediction of VO_2max_ a stepwise regression (backward elimination) was computed with inclusion of all the variables in the study (sex, body mass, height, BMI, and HRR). The final model retained only significant predictors (*p* < 0.05).

The validation process included three stages: (i) mean data comparison between the measured and estimated VO_2max_, (ii) simple linear regression, and (iii) Bland–Altman analysis. For the mean data comparison, the paired samples *t*-test (*p* < 0.05) between the measured and estimated VO_2max_ was computed. The mean difference, 95% confidence intervals, and Cohen’s d as effect size index were used.

Cohen’s d was deemed as: (i) trivial, if 0 ≤ d < 0.20; (ii) small, if 0.20 ≤ d < 0.60; (iii) moderate, if 0.60 ≤ d < 1.20; (iv) large, if 1.20 ≤ d < 2.00; (v) very large, if 2.00 ≤ d < 4.00; (vi) nearly perfect, if d ≥ 4.00 [32]. Simple linear regression between the VO_2max_ measured and estimated was computed. The trendline, determination coefficient (R^2^), standard error of estimation (SEE), 95% of confidence (95CI) and prediction (95PI) intervals were calculated. The qualitative interpretation of the relationship was defined as: (i) very weak, if R^2^ < 0.04; (ii) weak, if 0.04 ≤ R^2^ < 0.16; (iii) moderate, if 0.16 ≤ R^2^ < 0.49; (iv) high, if 0.49 ≤ R^2^ 0.8; and (v) very high, if 0.81 ≤ R^2^ < 1.0 [33]. Bland–Altman analysis included the plots of the difference and average of the VO_2max_ measured and estimated [34]. For qualitative assessment it was considered that at least 80% of the plots were within the ±1.96 standard deviation of the difference (95CI).

## 3. Results

The multiple linear regression retained, as significant, these VO_2max_ predictors: sex (*p* < 0.001) and HRR_1min_ (*p* = 0.003). The age, body mass, height, and BMI were not significant, in this model.

The prediction equation (R^2^ = 63.0%, SEE = 5.58) revealed a close relationship between measurements and can be expressed as:(1)$${\mathrm{VO}}_{2\max}=22.246+0.343\cdot \left({\mathrm{HRR}\_}_{1\min}\right)+11.722\cdot \left(\mathrm{sex}\right)$$
where VO_2max_ corresponds to the maximum oxygen uptake (mL/kg/min), HRR_1min_ corresponds to heart rate recovery (beats per minute) for one minute immediately after finishing the step test, sex corresponds to the value of zero for women and 1 for men. Table 2 presents the comparison between measured and estimated VO_2max_. The results showed non-significant differences with a trivial effect size.

Figure 1 presents the simple linear regression (panel A) between both variables. This revealed a close relationship (R^2^ = 63.3%, *p* < 0.001, SEE = 5.22). Bland–Altman plots (panel B) also fulfilled the agreement criteria, where more than 80% of the plots were within 95CI. In this case, all plots were within the 95CI.

## 4. Discussion

Our study confirmed the StepTest4all as a valid procedure in estimating VO_2max_ in the young adult population. Consequently, it was found that the magnitude of HR decrease immediately after exercise can be used to assess cardiovascular capacity. This procedure may be suitable for monitoring the level of cardiovascular capacity from an individual point of view, especially over time. This can be done by performing the StepTest4all periodically. However, some caution is needed when comparing VO_2max_ between different people. The same VO_2max_ value can mean different levels of physical capacity for a young person, an adult, a male, or a female. Thus, individual values must be checked against benchmark tables available in the literature [35] to see if they match the recommendations.

This test does not measure the performance in going up and down the step but reflects the cardiac recovery capacity after effort. The rise and drop of HR are controlled mainly by the sympathetic and parasympathetic nervous systems. Evaluating the magnitude of the decrease after exertion can also provide an assessment of nervous system function [36]. It is known that too small of a decrease in HR during the minutes following exertion cessation is associated with an increased likelihood of cardiovascular problems [37] and is even associated with premature death [7]. Conversely, a more rapid HR reduction after exercise is associated with a greater cardiovascular capacity [38]. However, there is still no consensus on the threshold value associated with a high risk of cardiovascular disease, the minimum value associated with an acceptable cardiovascular capacity, or nervous system dysfunction. For instance, Adabag et al. [39] in an article review, refer to values of 12–13 bpm of HRR in 1 min as threshold values. However, given the wide diversity of tests used and level of demand, some care is needed in defining the cut-off values (between normal and abnormal). As for the optimal recovery values, we know that healthy athletes can recover 60 or more bpm in one minute. Thus, the quality of recovery can be benchmarked with values between 12 and 60 bpm. Values close to 12 bpm may warn of a higher risk of cardiovascular disease or parasympathetic nervous system dysfunction; progressively higher values reflect very good cardiovascular capacity and good autonomic nervous system function [39,40]. The average HRR values found in our participants was 39 ± 10 bpm. This value is well above the minimum values aforementioned. Thus, they seem to agree with their age group and active lifestyle.

To provide a qualitative classification of cardiovascular capacity in this population group (young adults), we subdivided the range of HRR variation (12 to 60 bpm of recovery over one minute) usually found in these people and defined four categories, as can be observed in Table 3. This table shows a cardiovascular capacity classification and the reference values for VO_2max_ of participants between 20 and 29 years of age, obtained in our study, and the values of similar categories proposed by McArdle et al. [41]. Although the values are not completely coincident for men, in the case of women they are very close. This fact may mean that the values of VO_2max_ estimated by equation 1 are not discrepant in relation to other estimates.

In this regard, our experience with the StepTest4all allowed us to see that those values lower than 20 were usually associated with sedentary lifestyles and the presence of other risk factors (such as smoking and obesity) and values higher than 50 were usually present in people with high daily physical activity and a healthy lifestyle.

Because of its characteristics, the StepTest4all can be applied to people with different levels of physical ability and different somatic characteristics. To the best of our knowledge, this is the first step test that calculates the step height from several variables determining cardiovascular capacity. This calculation allows a primary adjustment, so that the test does not become too difficult/short or too easy/long. The finer adjustment is made by the precise control of the pace and its increment during the whole test. In this way, it is possible to reach the intensity of effort associated with 80% of HR_max_ (upper limit) for all in an adequate period (4 to 10 min).

In our study, age was an excluded variable from the VO_2max_ prediction model. This occurrence may be because our sample was made up of young adults. Probably, in samples with a larger range, age will also be part of the equation. The use of step tests, in addition to the aforementioned advantages, is appropriate because it is simple to apply, requires little space or means to carry out and can even be performed at home by anyone (see: https://www.facebook.com/StepTest4all (accessed on 28 July 2022)) [43]. The present study has some limitations: (i) it was carried out with a sample restricted to young adults, (ii) the sample was relatively small, and (iii) the amount and intensity of physical activity was not rigorously accounted for and did not enter the multiple linear regression as a possible factor. However, regarding this aspect, we can say that most participants in the sample had a similar weekly physical activity. Thus, further studies should be conducted to overcome these limitations. It should also be mentioned that the VO_2max_ and HR_max_ values of the participants were obtained by estimation. Notwithstanding, it was an estimation based on the evolution of the real individual values of VO_2_ and HR, both at rest and at different intensity levels. As mentioned earlier, this is a standard procedure generally used in non-athletic participants or in special population groups that should not be submitted to maximal test to exhaustion.

## 5. Conclusions

The StepTest4all was shown to be an adequate procedure to estimate cardiovascular capacity, expressed by VO_2max_, in young adults. The validation procedure indicated a high-level of agreement between the VO_2_ measured and estimated. Additionally, it is possible to determine the qualitative level of cardiovascular capacity from the HRR_1min_ more specifically, poor: <20 bpm, moderate: 20 to 34 bpm, good: 35 to 49 bpm, and excellent: ≥50 bpm. This procedure has the benefit of being simple to apply and can be used by everyone, even at home, without specialist supervision.

## Figures and Tables

**Figure 1 ijerph-19-11274-f001:**
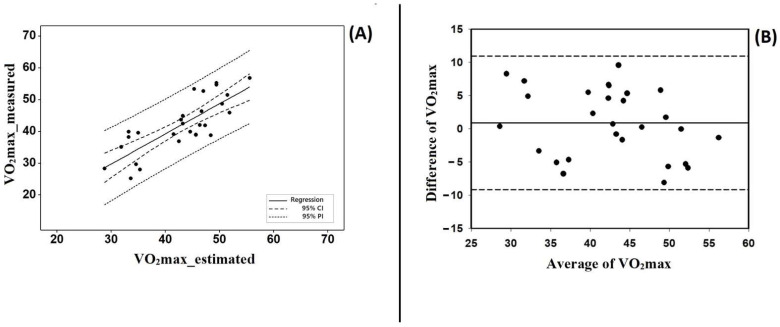
Linear regression between VO_2max_ measured and estimated by equation 1 (panel (**A**)), and Bland–Altman plots (panel (**B**)). All plots were within 95CI. This data refers to the validation group.

**Table 1 ijerph-19-11274-t001:** Participants’ characteristics. Equation group—group for VO_2max_ prediction equation and Validation group—group for validation of prediction model.

	Equation Group			Validation Group		
		Mean ± 1SD			Mean ± 1SD	
	Women (n = 10)	Men (n =18)	Total EG	Women (n = 9)	Men (n = 19)	Total VG
**Age [years]**	21.60 ± 1.84	21.17 ± 1.98	21.32 ± 1.90	20.44 ± 2.88	22.89 ± 3.80	22.11 ± 3.67
**Body mass [kg]**	61.45 ± 5.06	72.06 ± 8.83	68.27 ± 9.19	53.11 ± 5.42	72.92 ± 6.75	66.55 ± 13.31
**Height [m]**	1.63 ± 0.04	1.77 ± 0.07	1.72 ± 0.09	1.60 ± 0.04	1.75 ± 0.07	1.71 ± 0.10
**BMI [kg/m^2^]**	23.13 ± 3.02	22.93 ± 2.22	23.00 ± 2.48	20.75 ± 2.38	23.72 ± 2.11	22.77 ± 2.58
**HRR_1min_ [bpm]**	35.50 ± 10.23	40.22 ± 10.99	38.54 ± 10.79	34.89 ± 10.71	38.42 ± 10.42	37.29 ± 10.45
**VO_2max_ [mL/kg/min]**	34.41 ± 6.39	47.75 ± 6.71	42.99 ± 9.18	33.51 ± 5.64	46.15 ± 6.23	42.09 ± 8.45
**HRR_rest [bpm]**	81.90 ± 12.00	69.11 ± 10,94	73.68 ± 12.75	80.76 ± 10.60	64.37 ± 10.88	69.64 ± 13.16
**VO_2_rest_ [mL/kg/min]**	3.60 ± 0.86	3.61 ± 0.54	3.61 ± 0.66	3,54 ± 0.44	3.66 ± 0.64	3.62 ± 0.58

**Table 2 ijerph-19-11274-t002:** *t*-test paired samples comparison between the VO_2max_ measured and estimated in validation group. Effect size index (Cohen’s d) is also presented.

VO_2max_ Measured [mL/kg/min]	VO_2max_ Estimated [mL/kg/min]			
Mean ± 1SD	Mean ± 1SD	Mean difference (95CI)	*t*-test (*p* value)	d [descriptor]
42.09 ± 8.45	42.99 ± 7.10	−0.899 (−2.889 to 1.089)	−0.928 (0.362)	0.12 [trivial]

VO_2max_—maximal oxygen uptake.

**Table 3 ijerph-19-11274-t003:** Cardiovascular capacity (CVC) classification based on HRR_1min_ and the respective HRR cut-off values. It also presents the VO_2max_ values predicted by equation 1 (from our study) and the values proposed by McArdle et al. [41] for similar categories. These values are also used by a company of worldwide repute in the evaluation of body composition (Tanita: https://tanita.eu/blog/could-improving-your-vo2-max-be-the-secret-of-success (accessed on 28 July 2022)) [42]. These reference values are for the age group 20 to 29 years.

CVC_Classification	HRR_1min_	Men		Women	
		VO_2max_	McArdle et al., 2003 [42]	VO_2max_	McArdle et al., 2003 [42]
**Poor**	<20	<40	<36.5	<28	<29
**Moderate**	20–34	42–44.2	36.5–42.4	28–32.2	29–32
**Good**	35–49	44.3–49	42.5–46.4	32.3–36.9	33–36
**Excellent**	≥50	≥49	≥46.5	≥37	≥37

## Data Availability

Data is available under request to the contact author.

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
