# Peer review of "Validation of StepTest4all for Assessing Cardiovascular Capacity in Young Adults"

_ijerph, 2022, doi:10.3390/ijerph191811274_

Round 1
Reviewer 1 Report
The authors validated the StepTest4all as a procedure to estimate cardiovascular capacity in young adults, and found that Steptest4all is adequate to predict cardiovascular capacity. This is an interesting study, and the manuscript is well written. My concerns are listed below:
1. As the most important concept in the manuscript, StepTest4all was not well described. What does the StepTeat4all stand for? The authors stated that they developed the test. However, whether this method has been tested in other publications was not described. How reliable of the test? More information should be provided.
2. Since HRmax and VO2max in the manuscript were all estimated but not detected, I do not think it is okay to regress VO2max using other parameters. VO2max should be tested using the standard method.
3. The study was too simple to draw the conclusion. Only a regression was presented as the only evidence.
Author Response
RV_1
The authors validated the StepTest4all as a procedure to estimate cardiovascular capacity in young adults, and found that Steptest4all is adequate to predict cardiovascular capacity. This is an interesting study, and the manuscript is well written. My concerns are listed below:
As the most important concept in the manuscript, StepTest4all was not well described. What does the StepTeat4all stand for? The authors stated that they developed the test. However, whether this method has been tested in other publications was not described. How reliable of the test? More information should be provided.
Authors: First of all, thank you for your very pertinent comments and encouragement! Your suggestions are very clarifying, which greatly contributes to improving the quality of this manuscript. We have revised this manuscript according to tour suggestions.
Regarding the first comment, we have made some changes in the "data collection" we have improved the description of the test in order to make the description of the protocol clearer. We created subsection 2.3.
As we mentioned in the introduction, the existing step test protocols have some limitations. Therefore, we propose a test that solves the limitations of the previous tests. What we have developed is a step test protocol, which we call "steptest4all" that can be used with all kinds of people and that does not have the limitations of the previous ones.
StepTest4all has not been used before because it has not been validated yet. With this study we intend to validate this test, so that it can then be used with greater confidence.
Since HRmax and VO2max in the manuscript were all estimated but not detected, I do not think it is okay to regress VO2max using other parameters. VO2max should be tested using the standard method.
Authors: We agree with you about the ideal procedure for determining VO2max and HRmax - performing a progressive test to exhaustion, and taking into account objective criteria for accepting the VO2max value (e.g. VO2max plateau in the last legs of the test; HR close to the value obtained in the estimation from formulas, blood lactate above 8 mmol/L, or by observing fatigue symptoms).
When designing the study, we thought about the ideal procedure and what is possible in sedentary or active subjects, non-athletes, and therefore unfamiliar with testing to exhaustion. We weighed all the factors and concluded that the results we would obtain by performing a maximal test on these subjects would probably not be very reliable and it would be very difficult to reach all the validation criteria for the VO2max value. In addition, it would be necessary for the participants to be available to come to the laboratory several times because it would be necessary to train and perform the maximal test on another ergometer (e.g., treadmill or stationary bike). On the other hand, we also know that it is advisable to perform maximal tests in non-athletes, with medical supervision, even in apparently healthy subjects.
So, we searched for the most rigorous procedures to estimate VO2max. Among the multitude of ways to estimate it, we found that the best way is to use a submaximal, individual test, using the values of HR and VO2 at rest plus the values obtained in 3 or more submaximal steps. This way we obtained an individual regression that allows us to estimate the VO2max from the HRmax value. The direct and proportional association between the evolution of HR and VO2 has long been accepted. We clarified in the manunscript.
Among the various ways to estimate HRmax we chose the formula of Tanaka et al. (2001) - HRMax=208-0.7 x age, because it is the formula that was obtained from a large sample. The other formula (HRMax = 220 - age), although widely used to our knowledge has never been confirmed / validated by any study with large samples.
In a few words, to study these variables in this type of sample the procedure used was the one that we believe to be the most appropriate and possible.
Anyway, we appreciate the comment and will add this limitation to the study (see manuscript).
Regarding the quality of writing in English, the text was revised by a more qualified colleague who made minor changes.
- The study was too simple to draw the conclusion. Only a regression was presented as the only evidence.
Authors: In our study we used the standard validation procedures: i) T-Test to compare the predicted values with the actual values (see table 2); ii) The linear regression (figure 1, panel A); iii) Altman plots, with the confidence interval (figure 1 panel B). All procedures corroborate the validity of the estimate. We made minor adjustments for clarity’s sake.
Regarding the quality of writing in English, the text was revised by a native English speaker who made minor changes.
Reviewer 2 Report
In this work, authors purpose a study to validate a specific “Step Test” (StepTest4all) as an adequate procedure to estimate cardiovascular capacity in young adults. Subjects performed the StepTest4all protocol and were included in one of the following groups: (i) the group used to predict the maximal oxygen uptake; and (ii) the group to validate the prediction model. The validation process included three stages: (i) mean data comparison; (ii) simple linear regression, and (iii) Bland Altman analysis.
In general, the manuscript was well written with a detailed methodology and result sections.
Comments
Although the validation based on the mean data comparison, simple linear regression, and Bland Altman analysis was acceptable, I have some observations:
Really, the VOmax values were not measured (Table 2, Figure 1). I understand that this parameter was estimated using measured values of VO2 and HR, and also using the estimated value of HRmax. Therefore, values of VOmax in Table 2 and Figure 1 were both estimated values.
There is not information about how well the adjustment of the regression obtained with HR and VO2 was.
About the step test4all: i) how do you validated the formula for obtaining the height of the steps? why does that combination and no other?; ii) why does the cadence rises 2.5 cycles per minute and no 3, 2 or other?
Other:
Line 31 - To eliminate “in”
Line 139 – to rewrite “calculated considered”
Line 299 – there is not a verb in the phrase
Author Response
RV_2
Although the validation based on the mean data comparison, simple linear regression, and Bland Altman analysis was acceptable, I have some observations:
Really, the VOmax values were not measured (Table 2, Figure 1). I understand that this parameter was estimated using measured values of VO2 and HR, and also using the estimated value of HRmax. Therefore, values of VOmax in Table 2 and Figure 1 were both estimated values.
Authors: We agree with you about the ideal procedure for determining VO2max and HRmax - performing a progressive test to exhaustion, and taking into account objective criteria for accepting the VO2max value (e.g. VO2max plateau in the last legs of the test; HR close to the value obtained in the estimation from formulas, blood lactate above 8 mmol/L, or by observing fatigue symptoms).
When designing the study, we thought about the ideal procedure and what is possible in sedentary or active subjects, non-athletes, and therefore unfamiliar with testing to exhaustion. We weighed all the factors and concluded that the results we would obtain by performing a maximal test on these subjects would probably not be very reliable and it would be very difficult to reach all the validation criteria for the VO2max value. In addition, it would be necessary for the participants to be available to come to the laboratory several times because it would be necessary to train and perform the maximal test on another ergometer (e.g., treadmill or stationary bike). On the other hand, we also know that it is advisable to perform maximal tests in non-athletes, with medical supervision, even in apparently healthy subjects.
So, we searched for the most rigorous procedures to estimate VO2max. Among the multitude of ways to estimate it, we found that the best way is to use a submaximal, individual test, using the values of HR and VO2 at rest plus the values obtained in 3 or more submaximal steps. This way we obtained an individual regression that allows us to estimate the VO2max from the HRmax value. The direct and proportional association between the evolution of HR and VO2 has long been accepted. We clarified in the manunscript.
Among the various ways to estimate HRmax we chose the formula of Tanaka et al. (2001) - HRMax=208-0.7 x age, because it is the formula that was obtained from a large sample. The other formula (HRMax = 220 - age), although widely used to our knowledge has never been confirmed / validated by any study with large samples.
In a few words, to study these variables in this type of sample the procedure used was the one that we believe to be the most appropriate and possible.
Anyway, we appreciate the comment and will add this limitation to the study (see manuscript). Regarding the quality of writing in English, the text was revised by a more qualified colleague who made minor changes.
There is not information about how well the adjustment of the regression obtained with HR and VO2 was.
Authors: The determination of the "real" VO2max was done from the individual regression line of the values of VO2 and HR, at different levels and at rest. The figure below shows a typical example - See word file atached. The correlation is very high, usually with R2 greater than 0.97. In the example below even higher. In the manuscript we added the range of the regression (R2) between the HR-VO2 assessed for each participant. We clarified in the manuscript.
About the step test4all: i) how do you validated the formula for obtaining the height of the steps? why does that combination and no other?; ii) why does the cadence rises 2.5 cycles per minute and no 3, 2 or other?
Authors: Thank you very much for the question. it is a very pertinent question on which we have reflected in great detail. The formula was not "statistically" validated but was "validated" experimentally. That is, in the course of carrying out dozens of tests, we verified that a step height between 15 cm and 40 cm was sufficient to achieve the test objectives in a reasonable time (5 to 10 minutes), any type of subject (from subject with very poor physical ability to long-distance runners).
A height of 40cm has been used in other protocols. It is not common to use heights greater than 40 cm. The height of 40 cm is a very demanding height, even for subjects with good physical capacity and tall stature.
Thus, we introduced variables in the formula until we reached the one presented in the manuscript). Why, in our view, does this formula for determining the height of the step not need special validation? Because the height of the step is only one of the components of the protocol and although very important, the very precise determination is not critical, as it can be compensated by adjusting the rhythm of ascent and descent, controlled by the metronome. Thus, a small "failure" in the height of the step is compensated for by carrying out one or two more steps. The achievement of more or less one level does not invalidate the achievement of the test objective - reaching 80% of the estimated HRmax in 5 to 10 minutes.
In short, the formula that gives us the height of the step, even with some slight inaccuracy, is later compensated by adjusting the pace, until reaching the test objective. The dozens of tests carried out by us, using that formula, never prevented the achievement of the intended final intensity.
As for the rhythm, why increase 2.5 cycles per minute? We thank you for the question, which is very timely and which was the subject of our reflection. The choice of this value took into account the fact that, with this magnitude of increase (and starting from a base rate of 15 cycles per minute, at level 1), in the event that the participant takes 10 minutes to perform the test, the value of the rhythm at the last level would be 37.5 cycles per minute. This pace is very difficult to be performed even by someone with good ability, as it would have to be done as if it were in a race. In addition to being difficult to implement the final rhythm, it would cause inaccuracies in the performance of the movement. Thus, the limits of 15 cycles (very slow pace that can be performed by anyone and also serve as a warm-up) and 37.5 cycles per minute were what we considered adequate and possible.
Other:
Line 31 - To eliminate “in”
Authors: We made the changes in manuscript
Line 139 – to rewrite “calculated considered”
Authors: We made the changes in manuscript
Line 299 – there is not a verb in the phrase
Authors: We made the changes in manuscript

Round 2
Reviewer 1 Report
Although the authors responded to my concerns and revised the manuscript accordingly, I suggest the authors to provide an extensive discussion of the limitation of the manuscript. Since my major concerns were not well addressed.
Author Response
Minor revisions
RV1 - Although the authors responded to my concerns and revised the manuscript accordingly, I suggest the authors to provide an extensive discussion of the limitation of the manuscript. Since my major concerns were not well addressed.
Authors: First, we would like to thank you for the careful review and the questions asked. Since the reviewer mentions that your concerns were not well addressed in revision 1, we will try to clarify each one in more detail. Please note that new changes are underlined blue.
In this second revision we are highlighting what the reviewer mention as limitation of the manuscript: first revision comment “Since HRmax and VO2max in the manuscript were all estimated but not detected, I don't think it is correct to regress VO2max using other parameters. VO2max should be tested using the standard method”. We also made some improvements in the protocol description.
In the first review we tried to clearly justify why we estimated HRmax and VO2max. In addition, we can say that with the subjects in this sample, it would not be appropriate to perform maximal testing to exhaustion. They are sedentary or active people. Our experience performing maximal tests with athletes and other subjects allows us to say that the criteria for reaching VO2max when testing sedentary people to exhaustion are rarely reached. We made some changes in the manuscript.
On the other hand, as mentioned in the answer to revision 1, the individual regression line, which associates HR-VO2 showed R2 always higher than 0.96 and in many cases higher than 0.99. This finding of such a strong association between these two variables is described in exercise physiology textbooks (e.g. McArdle, W. D., Katch, F. I., & Katch, V. L. (2010). Exercise physiology: nutrition, energy, and human performance. Lippincott Williams & Wilkins).
Thus, we agree with the reviewer's opinion that the ideal was to perform maximal tests, but in the case of the present study, we argue that the procedure used is the most appropriate, given the characteristics of our sample. The estimation performed is the one adequate way to study specific population groups or elderly people. Our experience performing maximal tests with athletes and other subjects allows us to say that the criteria for reaching VO2max, when testing sedentary people to exhaustion, are rarely reached.
In any case, we have added this justification in the manuscript.
In the response to revision 1 we significantly changed the limitations section of the study. In this second revision we hope that we have justified the notions made by amending the manuscript with more detailed information that justifies the procedures used.
However, it is curious and interesting to note that reviewer 1 in the first review stated that: “Is the research design appropriate?” and “Are the methods adequately described?” has set "yes" and in the second review, after improving the manuscript, has changed it to "can be improved". This leaves us a little confused and with more difficulty in responding to reviewer 1